# Oxidative and Glycation Damage to Mitochondrial DNA and Plastid DNA during Plant Development

**DOI:** 10.3390/antiox12040891

**Published:** 2023-04-06

**Authors:** Diwaker Tripathi, Delene J. Oldenburg, Arnold J. Bendich

**Affiliations:** Department of Biology, University of Washington, Seattle, WA 98195, USA; tripad@uw.edu (D.T.); delene@uw.edu (D.J.O.)

**Keywords:** AGEs, antioxidants, DNA repair, maize, mtDNA, ptDNA, ROS

## Abstract

Oxidative damage to plant proteins, lipids, and DNA caused by reactive oxygen species (ROS) has long been studied. The damaging effects of reactive carbonyl groups (glycation damage) to plant proteins and lipids have also been extensively studied, but only recently has glycation damage to the DNA in plant mitochondria and plastids been reported. Here, we review data on organellar DNA maintenance after damage from ROS and glycation. Our focus is maize, where tissues representing the entire range of leaf development are readily obtained, from slow-growing cells in the basal meristem, containing immature organelles with pristine DNA, to fast-growing leaf cells, containing mature organelles with highly-fragmented DNA. The relative contributions to DNA damage from oxidation and glycation are not known. However, the changing patterns of damage and damage-defense during leaf development indicate tight coordination of responses to oxidation and glycation events. Future efforts should be directed at the mechanism by which this coordination is achieved.

## 1. Introduction

In order for an organism to survive and reproduce, its chromosomal DNA must be maintained and faithfully replicated. DNA molecules, however, are susceptible to damage of two types. Endogenous damage occurs when DNA polymerase makes mistakes [1], and when metabolic byproducts react with the DNA, either oxidatively or by glycation reactions [2]. Exogenous damage can be caused by radiation and toxins [3]. Both types of damage affect the genomes in all organisms, but plant cells carry three distinct genomes (nuclear, mitochondrial, and plastid), so that damage and its repair may differ among the three. As described below, recent research has shown that damage (per kb of DNA) to the genomes in the cytoplasmic organelles (organellar DNA, or orgDNA) is greater than that in the nucleus, and that damage to orgDNA increases as the promitochondria and proplastids in the meristem develop into mature organelles in green leaves. Furthermore, the biochemical “effort” to repair damage to orgDNA molecules appears to decrease as the plant develops from meristem to green leaf.

Most orgDNA damage/repair research has focused on a specific stage of plant development (e.g., the entire plant or the leaf) in the presence or absence of a perturbation (e.g., drought, radiation, mutation, pathogen). Here, we review recent data on damage/repair during the normal development of maize (*Zea mays* L.) seedlings in the absence of a perturbation. We consider endogenous orgDNA damage caused by oxidation and glycation, both of which, surprisingly, follow the same pattern during development from meristem to leaf. The demise of maize orgDNA during development may be attributed to oxidative and glycation damage that is not repaired.

## 2. Cellular Metabolism and ROS

Approximately 2.7 billion years ago, photosynthetic organisms introduced molecular oxygen into earth’s early reducing atmosphere, driving the origin of reactive oxygen species (ROS, including ^1^O_2_, H_2_O_2_, O_2_^•−^, and OH^•^ radicals) [4]. As unavoidable byproducts of electron transport during respiration and photosynthesis, ROS are continuously produced by aerobic metabolism within cell compartments (chloroplasts, mitochondria, and peroxisomes) and the apoplast [5,6,7]. Low levels of ROS can benefit the plant by serving an important cellular function in communication between the nucleus and the organelles, and maintenance of cellular homeostasis [8,9,10]. On the other hand, high levels of ROS molecules can lead to oxidative stress and cause extensive damage to DNA, protein, and lipids, and, if not promptly repaired, may kill the plant cell [9,11,12,13].

Plants have various systems to deal with potential oxidation damage, and to maintain cellular redox homeostasis [14,15,16]. The antioxidant machinery includes proteins such as superoxide dismutase (SOD), ascorbate peroxidase (APX), guaiacol peroxidase, glutathione-S-transferase, and catalase (CAT), as well as small molecules such as ascorbic acid (AsA) and reduced glutathione (GSH) [17,18,19]. Many of these components reside within organelles where respiration and photosynthesis generate ROS; plastids also contain antioxidant pigments such as carotenoids [9]. However, the genes for organellar-targeted antioxidant components are all nuclear-encoded and likely regulated by nuclear–organellar crosstalk and redox metabolism [20,21].

### 2.1. ROS during Abiotic and Biotic Stresses

Compared with most animals, sessile plants are more frequently subjected to changing environments and abiotic stress factors such as extreme temperature changes, drought, and pollutants. Additionally, biotic stress may arise when energy demands change, especially during the transition from non-green to green cells and under high-light conditions [22]. Both abiotic and biotic factors influence cellular homeostasis and ROS production [23]. Excess ROS can result in protein and DNA damage and initiation of global damage response mechanisms, similar to those observed in bacteria [15,24]. Although the various mechanisms by which plants respond to changing levels of ROS are beyond the scope of this review, we emphasize two points: (1) maintaining low ROS (low respiration metabolism and no photosynthesis) protects against DNA damage; and (2) the levels of proteins and small molecules that replicate and defend orgDNA are probably controlled by nuclear-encoded, ROS-responsive genes.

ROS are formed during the partial reduction of molecular oxygen. Therefore, maintaining certain cells under hypoxic conditions can prevent ROS from causing damage. In both mammalian and plant cells, hypoxia is maintained in cells destined to form gametes, thus protecting the DNA from potential oxidative damage during transmission to the next generation [25,26,27]. In Arabidopsis (*Arabidopsis thaliana* L.), maintaining mitochondrial function within the shoot apical meristem and root apical meristem is vital to maintaining stem cell activity and adapting to temperature stress throughout development [28]. Furthermore, the non-green cells of Arabidopsis are maintained in a hypoxic niche for five weeks during the development of the inflorescence meristem of the adult plant [29]. Thus, the DNA to be transmitted to the next generation is protected from potential oxidative stress associated with respiration and photosynthesis [25,26]. In addition, anti-ROS enzymes and small antioxidant molecules can modulate oxidative stress, and signal the redox status of the cell to the nucleus, possibly by setting the level of H_2_O_2_, the most highly mobile of the various forms of ROS [30,31,32,33].

### 2.2. ROS-Targeted Damage to Proteins and DNA

ROS production can result in the oxidation of proteins, and likely is greater during conditions of stress. Proteins undergo different types of modifications, either directly or indirectly. Chemical modifications such as nitrosylation, carboxylation, disulfide bond formation, and glutathionylation can directly change protein activity. Protein carbonylation is often used to evaluate protein oxidation [34]. As a result of interaction with lipid peroxidation products, proteins can undergo indirect modification. Once ROS concentrations reach a threshold, amino acids undergo site-specific modifications that lead to proteolytic degradation [34].

Due to their proximity to the sites of membrane-associated ROS during respiration and photosynthesis, mitochondrial DNA (mtDNA) and plastid DNA (ptDNA) are at higher risk of ROS attack than nuclear DNA. ROS-induced oxidation can lead to many types of DNA damage: modification of the nucleotide base; oxidation of the deoxyribose sugar residue; oxidation products including 8-hydroxyguanine (8-oxoG), hydroxyl methyl urea, dehydro-2-deoxyguanosine, and thymine glycol; abstraction of a nucleotide; single-strand breaks; and cross-linking of DNA with associated proteins [4,35]. Mitochondria and chloroplasts possess DNA repair pathways similar to most nuclear ones. Proteins associated with orgDNA repair are encoded exclusively by the nuclear genome [3,36,37]. These repair systems include base-excision repair (BER) in Arabidopsis plastids and mitochondria [3,38] and Arabidopsis organellar DNA polymerases, mediating microhomology-mediated end-joining [39]. The BER system is the major pathway for repair of oxidatively-damaged DNA [40]. As described below, the level of orgDNA repair/avoidance changes during development (high in meristem and low in green leaves), and is likely regulated by cellular redox status, especially by H_2_O_2_ levels. Further, if a DNA molecule carrying damage is not repaired, it will degrade to prevent mutagenesis, known as DNA abandonment [41,42,43].

### 2.3. Assessing ROS and Oxidative Damage to orgDNA in Maize

The molecular structure, size, and copy number of orgDNA change during maize development from multi-genomic branched forms in the meristem to subgenomic fragments in the green leaf [43,44,45,46]. In addition, dark-grown plants retain high-integrity orgDNA in the leaves, and in plastids, the ptDNA copy number decreases rapidly after transferring from dark to light growth conditions [47]. We attributed this loss of molecular integrity to ROS-induced DNA damage that was not repaired, followed by degradation of the damaged orgDNA [43].

Box 1 describes procedures used to quantify oxidative damage, and we have used some of these to study ROS and orgDNA damage as wild-type maize seedlings developed “normally” (without exposure to genotoxic agents or extreme environments) [27]. Maize was chosen because (i) monocots (such as maize) have a larger leaf meristem than do most dicots, including Arabidopsis and tobacco (*Nicotiana tabacum* L.), facilitating biochemical measurements at all stages of leaf development; (ii) cellular development progresses linearly from the meristem at the base of the leaf to the leaf blade, so that tissues representing stages of development are easily obtained; and (iii) orgDNA damage increases and its molecular integrity decreases during normal leaf development [43]. Changes in ROS and orgDNA damage were then correlated with prior data as to the molecular integrity of orgDNA in order to investigate possible cause-and-effect relationships (Table 1 and Figure 1). 

The levels of ROS (H_2_O_2_ and O_2_^−^) were lower in stalk tissues than in leaves, and ROS levels increased as the seedlings developed from stalk to leaf blade [27]. Antioxidative agents were also assayed. Cellular CAT and peroxidase activities decreased, whereas small antioxidants (GSH and AsA) increased. Plastids isolated from the lower part of the stalk contained pristine DNA molecules, whereas these molecules were highly degraded in green leaves [44,45,46]. 8-oxoG is an indicator of oxidative damage to DNA, and the production of ROS and 8-oxoG was found to increase during maize leaf development. We concluded that damaged-but-not-repaired orgDNA was degraded by default, as has been observed to occur in *Escherichia coli* [49]. 

Studies of light-grown versus dark-grown maize seedlings were also informative. Dark-grown leaves had lower levels of ROS, H_2_O_2_, superoxide, and SOD than light-grown leaves [27]. The amount and molecular integrity of ptDNA remained high in dark-grown leaves, but declined rapidly when maize seedlings were transferred from dark to light environments [47]. Although ROS generated in the dark as byproducts of respiration could lead to mtDNA damage, we speculate that a small amount of H_2_O_2_ signals the nucleus to express and deliver DNA-repair proteins to the mitochondria and plastids, so that high-integrity orgDNA is maintained in dark-grown leaves. In the light, the green cells in the leaf blade increase production of both ROS (leading to damaged orgDNA) and H_2_O_2_. Elevated levels of H_2_O_2_ from the chloroplasts reach the nucleus, signaling the nucleus to cut off the supply of DNA repair proteins (such as RecA), resulting in the disintegration of damaged DNA in both organelles.

By controlling the levels of oxygen and antioxidative agents, the redox status of cells can be adjusted to achieve an “objective” during plant development. Minimizing DNA damage in the small, slow-growing cells of the germline facilitates transmission of pristine orgDNA molecules during sexual reproduction, whereas ROS-generating respiration and photosynthesis facilitate rapid growth of leaf cells, even though damage to orgDNA increases.

## 3. Cellular Metabolism and Glycation

Photosynthesis produces sugars that can lead to glycation damage to both small and large molecules anywhere in the cell. Glycation involves the non-enzymatic covalent addition of a reactive carbonyl group from sugars or the small-molecule glycolytic byproducts, such as methylglyoxal [CH3–CO–CHO; MGO] and glyoxal [CHO–CHO; GO], to amino and thiol groups on proteins, lipids, and nucleic acids [50,51]. 

Glycation was discovered in 1912 when Louis Maillard observed browning in a heated solution containing sugar and an amino acid, and the physiological effects of reactive acyclic oxyaldehydes, such as MGO and GO, have been widely studied [52]. All organisms produce MGO spontaneously during glycolysis, using both enzymatic and non-enzymatic pathways. In plants, MGO is produced primarily during glycolysis and the Calvin cycle through non-enzymatic reactions. In contrast, GO formation occurs when monosaccharides, saccharide derivatives, and proteins are peroxidized and glycated. Through a series of intermediates, glycation finally results in stable end-stage adducts called advanced glycation end-products (AGEs) [52]. As described below, early-glycation adducts may be removed from small molecules, proteins, and DNA, but no process has been identified for removing AGEs. Thus, AGEs are probably the most important agents of glycation damage to cells [53,54].

There are several notable aspects of glycation damage to cells:Although the genotoxic effects of ROS have been intensively studied, most such reports do not mention glycation damage. Furthermore, the relative contributions to DNA damage from oxidation and glycation are not known.Defense against glycation damage to DNA involves the same enzymes that cleanse the nucleotide pool before polymerization (damage avoidance) and reverse the damage in DNA molecules (deglycase activity). That the structures of these enzymatic substrates are so different is remarkable.Perhaps most surprising is the tight coordination of defenses against DNA damage caused by oxidation and glycation.

DNA is susceptible to glycation by MGO and GO, and DNA damage due to glycation has been reported to be comparable to that due to ROS [53]. In physiological conditions, the most reactive nucleotide is deoxyguanosine (dG), and glycation can generate the GO–DNA adduct, dG-G, and the MGO–DNA adduct, dG-MG. DNA strand breaks, high mutation frequency, and cytotoxicity were observed in cells with these DNA modifications. The dG-MG adduct is five times more prevalent than the most common oxidative adduct (8-oxoG) among nucleotides in DNA from cultured human cells [52]. However, because cultured cells do not reflect the conditions experienced in intact tissues, and no other glycation/ROS comparisons have been reported, the relative contributions of glycation and oxidative damage to DNA in plants remain to be investigated.

As is the case with oxidative damage, organisms can avoid and reverse glycation damage. The glyoxalase pathway, which comprises glyoxalase I (GLY I), glyoxalase II (GLY II), and glyoxalase III (GLY III), is the main pathway for detoxifying MGO in all living organisms [55,56]. DJ-1, a GLY III family member, and its bacterial homologs can repair DNA and RNA containing MGO- or GO-glycated nucleotides. Moreover, depletion of DJ-1 is associated with increased glycated DNA, DNA strand breaks, and strong mutant phenotypes [53]. DJ-1, also known as Park7 in humans, maintains DNA integrity by repairing guanine glycation damage. DJ-1 also repairs proteins damaged by MGO- and GO-glycated cysteines, arginines, and lysines [57], and is the only known enzyme that can repair damage to both nucleic acids and proteins. DJ-1 can ameliorate damage to DNA in two ways: (i) by acting as a deglycase to reverse the damage in DNA; and (ii) by detoxifying the small-molecule aldehydes in a glyoxalase damage-avoidance process [58,59].

Arabidopsis contains three DJ-1 deglycase proteins that are 35–40% similar in amino acid sequence to human DJ-1/Park7 [60]. One of these, AtDJ-1a, promotes antioxidant activity by protecting the plant from intense light [60]. Chloroplasts and mitochondria contain AtDJ-1b, a sulfenylated protein with a redox-sensitive function [61]. In young leaves, AtDJ-1c is more abundant than in older leaves, which indicates it is essential for plastid development [62]. There are 11–12 genes for DJ-1 glyoxalase proteins in the rice and maize nuclear genomes, and some of these DJ-1 proteins are predicted to be dual-targeted to both mitochondria and plastids [63]. Although DJ-1 may protect against and repair early-stage glycation modifications, the end-stage AGE damage to proteins and DNA is irreversible, so that the degradation of these damaged molecules likely would be the best (or only) cellular-defense strategy.

### Assessing Glycation Damage to Maize Proteins and orgDNA

Plant glycation studies typically have focused on proteins. Using maize, Tripathi et al. studied glycation damage in organelles, orgDNA, and proteins during development under normal growth conditions [48]. MGO and GO are formed from the degradation of monosaccharides, glycated proteins, and glycolytic intermediates, and procedures used to quantify glycation are described in Box 2.

In both plastids and mitochondria, the level of free MGO was higher in green leaves than in the meristem and stalk tissues [48]. The early-stage Amadori modifications and late-stage AGE adducts in orgDNA, as well as AGE in proteins, were also found to increase with development. In contrast, levels of DJ-1 protein in both plastids and mitochondria decreased from the basal meristem to the mature leaves (Table 1 and Figure 1). In addition, lower DNA glycation damage and higher DJ-1 levels were found in dark-grown leaves compared to light-grown leaves. DJ-1 levels were higher in stalk than leaf and higher in organelles from dark-grown than light-grown leaves, suggesting DJ-1 provides protection from glycation damage. 

In summary, glycation damage and damage-defense follow a similar developmental pattern as ROS damage and damage-defense: as damage increases, damage-defense measures decrease. This surprising observation raises two questions: (1) Why abandon damaged orgDNA when the demand for orgDNA gene products (such as ribosomal RNA, cytochrome oxidase, and the psbA gene product, D1) increases to support seedling growth? (2) How does the cell accomplish the remarkable and nearly quantitative coordination of damage/damage-defense for glycation and ROS, and do this in both mitochondria and plastids?

## 4. Linkage of Damage Caused by Oxidation and Glycation

Over the years, an association between oxidative damage and glycation damage has been reported in diverse biochemical studies, as recounted here. In the presence of H_2_O_2_, MGO caused the conversion of 2′-deoxyguanosine to a DNA adduct, a N_2_-acetyl derivative of guanine (N_2_-acetyl-2′-deoxyguanosine), demonstrating their synergistic damaging effects on DNA [64]. Abordo et al. found that the cellular concentrations of GO, MGO, and 3-deoxyglucosone increased during oxidative stress, an effect attributed to the limited availability of reduced cofactors (GSH and NADPH) needed by glyoxalases for the enzymatic detoxification of glycated substrates [65]. Free radicals are generated at every step of the glycation process. For example, AGEs and Amadori products produce free radicals after they react with oxygen [66]. The glycation of proteins by MGO and GO also produces ROS (superoxide, hydroxyl radicals, and H_2_O_2_) [51]. Rosca et al. showed three findings: a link between the formation of intracellular AGEs on renal mitochondrial proteins and ROS; glycation-damaged mitochondria were functionally impaired due to increased superoxide and oxidative damage to mitochondrial proteins; and that, in chronic diabetes, proteins were prone to deleterious posttranslational modifications resulting from glycation and oxidation [67]. A similar link between glycation and oxidation was found in aging worms [68].

High levels of ROS generated in mitochondria can lead to increases in MGO, AGEs, and cell death [69]. Oxidative stress causes mtDNA glycation in cultured mouse cells deficient in superoxide dismutase [70]. When MGO reacts with lysine in vitro, H_2_O_2_ and superoxide are produced [71]. 

These reports establish a strong link between glycation and oxidation. In our studies, maize development was accompanied by increasing oxidative damage to mtDNA and ptDNA and increasing AGEs in orgDNAs (Table 1 and Figure 1). Taken together, plants and animals coordinate oxidative and glycation damage and damage response for their orgDNAs. 

## 5. Integrity of orgDNA Molecules during Maize Development

As mentioned above, the nucleus “senses” the redox status of the cell, regardless of where in the cell the signal (H_2_O_2_) is produced, and then dictates the fate of the orgDNA. In order to reduce the metabolic cost of DNA repair, germline cells are powered by “quiet” metabolism (neither respiration nor photosynthesis), and somatic cells by “active” metabolism (both respiration and photosynthesis) [41]. Although full repair of DNA damage is required in the meristem, low oxygen, low H_2_O_2_, high antioxidants, and high DJ-1 deglycase suppress oxidative and glycation damage, reducing the cost to repair both orgDNA and nuclear DNA. Repair pathways known to operate in plant organelles, including BER and homologous recombination [3,39], suffice to maintain pristine orgDNAs. Following ROS signaling to suppress organellar-targeted DNA-repair proteins in green maize leaves, orgDNA damage from ROS and glycation accumulates, leading to orgDNA fragmentation. Although the repair of nuclear DNA is needed for cellular homeostasis and checkpoint control of cell division, somatic leaf cells can “afford” not to repair their damaged organellar genomes, thus reducing the high cost of orgDNA repair.

## 6. Detection of Oxidative and Glycation Damages

### 6.1. Box 1. Measuring ROS, Oxidative Stress, and DNA Damage

Numerous methods have been employed to quantify ROS-associated components in bacterial, mammalian, and plant cells, and to assess changes in these components following normal cellular development or in response to stress [72]. Chemical probes that preferentially react with a specific ROS molecule are widely used for histochemical staining or fluorescent detection of ROS. Tetrazolium dyes such as nitro blue tetrazolium (NBT) and XTT reagent (2,3-bis-(2-methoxy-4-nitro-5-sulfophenyl)-2H-tetrazolium-5-carboxanilide) are frequently used in situ. Fluorescein and rhodamine dyes are fluorogenic probes for detecting oxidative activity in cells and tissues. For H_2_O_2_ detection, the fluorescent probe Amplex Red (N-acetyl-3,7-dihydroxyphenoxazine) and its derivatives are widely used. Fluorescent biosensors, nanoprobes, and Electron Paramagnetic Resonance spectroscopy are used to quantify oxidants [72,73,74]. For catalase (CAT) detection, commercial kits are based on the principle that CAT is quenched with sodium azide, and the remaining H_2_O_2_ facilitates the coupling of 4-aminophenazone (4-aminoantipyrene) and 3,5-dichloro-2-hydroxy-benzenesulfonic acid catalyzed by horseradish peroxidase. The product, quinoneimine dye, is measured using an absorbance assay [27].

Numerous methods are used to assess oxidative stress and ROS accumulation in plants. These include measuring changes in the levels of biochemicals such as chlorophyll, malondialdehyde, anthocyanin, proline, and glycine betaine [72]. Enzyme activity measurements for SOD, APX, CAT, glutathione reductase, peroxidases, dehydroascorbate reductase, and monodehydroascorbate reductase, as well as non-enzymatic assays (for AsA, GSH, vitamins, flavonoids, anthocyanins, phenolics, 2,2-diphenyl-1-picrylhydrazyl), are also used to assess ROS levels [75]. 

One way to assess DNA damage and repair following genotoxic exposure, or in mutants lacking repair functions, employs long-PCR to quantify DNA lesions [76]. This method has been used to show increased DNA damage in a DNA polymerase mutant of Arabidopsis [77] and greater orgDNA damage in light-grown maize seedlings compared to those grown in continuous dark [47]. A major type of oxidative DNA damage is 8-oxoG base modification [24,78,79], and this nucleoside adduct is commonly assayed using antibodies and either a competitive ELISA method or immunofluorescence microscopy [27,80].

### 6.2. Box 2. Measuring Glycating Agents and Damage to Proteins and DNA

Methods for measuring MGO and GO use derivatization with 1,2-diaminobenzene and detection by gas chromatography/mass spectrometry with stable isotopic dilution (GC–MS/MS), as well as enzymatic reactions of glutathione/glyoxalase I [51]. Assays of DNA glycation involve determining Amadori products in DNA using an NBT reduction assay [81,82]. Detection of MGO is achieved using a commercial methylglyoxal assay kit containing engineered enzymes and a chromophore (Abcam), which, in reduced form, is quantified [63]. Assessment of AGE products in organelles employs either a commercially-available AGE ELISA assay (Cell Biolabs) or a quantitative dot blot assay with AGE antibodies. For DJ-1 determination, assays include ELISA with commercial antibodies to human Park7 (Abcam) and a quantitative dot blot assay using DJ-1 antibodies [48].

## 7. Concluding Remarks

Plant cells carry many copies of orgDNA molecules, copy numbers change during development, and species differ in how they deal with the problem of orgDNA damage [43,83]. At one extreme, maize allows the orgDNA molecules to go unrepaired as leaves expand. However, in tobacco leaves, some ptDNA molecules appear to be intact [84], implying that repair is ongoing. Since orgDNA replication seems to be restricted to meristematic and developing cells [85], the “strategy” used by maize is to produce long-lasting RNA transcripts from intact orgDNA templates in the meristem, to maintain leaf respiration and photosynthesis for the single growing season. To what extent does the developmental transition from orgDNA protection to extensive ROS and glycation damage (Figure 1) apply to tobacco?

The repair of DNA damage in plant organelles appears to involve the coordination of defenses against oxidative and glycation damage, involving both small and large molecules. How might this be accomplished? The “SOS response” to DNA damage in bacteria involves a surprisingly simple repression/activation system to coordinate the expression of ~40 to 50 genes [86]. In plants, with three distinct genomes and many cell types, coordination might be expected to be more complex. However, the way in which a cell begins the stress-response may be similar among organisms. Since orgDNA is more susceptible to damage than nuclear DNA, and nearly all genes from the primordial endosymbionts have already moved to the nucleus, why has any orgDNA been retained? The answer provided by Wright et al. is that high vulnerability to damage allows orgDNA to serve as a sensor of redox damage and, we would add, glycation damage [87]. When DNA polymerase reaches a damaged site, it pauses, and the short single-stranded DNA (ssDNA) segment at the replication fork is quickly converted to double-stranded DNA after repair. However, if repair factors are unavailable, ssDNA persists, leading to aberrations (strand breaks, mutations, etc.). Persistent ssDNA presumably alters the redox status, and that signal (perhaps in the form of H_2_O_2_) reaches the nucleus.

This above scenario is speculative and can be tested in future research. We note, however, that, whereas oxidative damage to plant orgDNA has long been known [43,88], research on glycation damage has only just begun [48].

## Figures and Tables

**Figure 1 antioxidants-12-00891-f001:**
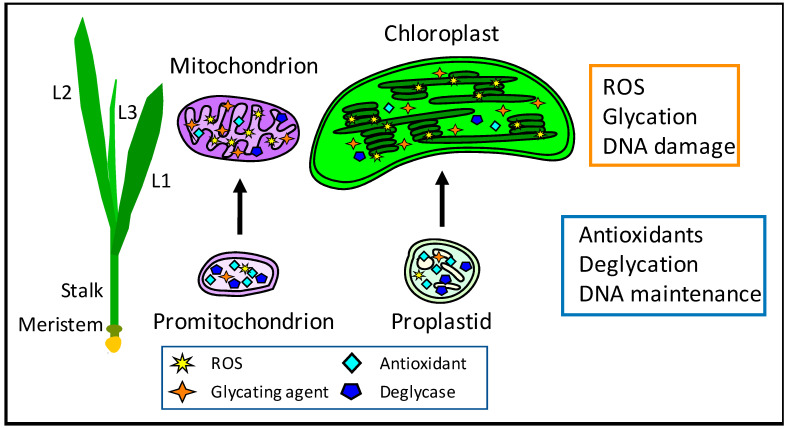
Damage in maize organelles during development. Cellular and organellar development in maize proceeds from the basal meristem to the fully expanded leaf blade. For light-grown maize seedlings, samples from lower 1/3 stalk (includes basal meristem), upper 2/3 stalk, and first leaf (L1) were assayed for ROS, glycation, and orgDNA damage [27,48]. Similar assays were performed for seedlings grown in light or in continuous dark using tissue from the first three leaf blades (L1, L2, and L3). Levels of antioxidants and the deglycase DJ-1 protein were higher in promitochondria and proplastids than in mature organelles. In contrast, levels of ROS and glycation products were higher in mature mitochondria and chloroplasts, where greater orgDNA damage was evident. Less ROS, glycation, and DNA damage were also found in dark-grown, etiolated leaves than in leaves grown under normal light conditions. The developmental changes from ROS and glycation damage to orgDNA result in a lower number of functional genome copies among the number typically scored as “copies” using methods such as standard qPCR [27,45,46,48].

**Table 1 antioxidants-12-00891-t001:** Summary of ROS and glycation assays in maize.

**ROS Assays ^1^**	**Tissue**	**Condition**
General ROS	Leaf > stalk ^2^	Light > dark ^3^
Hydrogen peroxide	Leaf > stalk	Light > dark
Superoxide ^4^	Leaf > stalk	Light > dark
Superoxide dismutase	Leaf > stalk	Light > dark
Catalase	Leaf < stalk	Light < dark
Peroxidase	Leaf < stalk	Light < dark
Glutathione	Leaf < stalk	Light < dark
Ascorbic acid	Leaf < stalk	Light < dark
8-oxoG in DNA	Leaf > stalk	Light > dark
**Glycation assays ^5^**		
MGO	Leaf > stalk	Light > dark
AGEs in proteins	Leaf > stalk	Light > dark
Amadori-adducts in DNA	Leaf > stalk	Light > dark
AGEs in DNA	Leaf > stalk	Light > dark
DJ-1	Leaf < stalk	Light < dark

^1^ Results of ROS assays for plastids and mitochondria in maize, with the following exceptions: data for catalase, glutathione, and ascorbic acid were from protoplasts. Data from Tripathi et al., 2020 [27]. ^2^ Maize seedlings were grown under normal 16-h light/8-h dark conditions. Stalk includes two tissue sections: lower 1/3 of stalk and upper 2/3 of stalk, composed of basal meristem and developing cells. Leaf is the green, fully-expanded first leaf blade. ^3^ Maize seedlings were grown under 16-h light/8-h dark conditions or continuous dark. The tissue from the first three leaves was used. ^4^ Superoxide assay was only performed with mitochondria, not with plastids. ^5^ Results of glycation assays for plastids and mitochondria in maize. Data from Tripathi et al., 2022 [48]. 8-oxoG: 8-hydroxguanine, MGO: methylglyoxal, AGEs: advanced glycation end-products.

## Data Availability

No new data were created or analyzed in this study. Data sharing is not applicable to this article.

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
