# Peer review of "Oxidative and Glycation Damage to Mitochondrial DNA and Plastid DNA during Plant Development"

_antioxidants, 2023, doi:10.3390/antiox12040891_

Round 1

Reviewer 1 Report

The manuscript is quite interesting. The English form is good. The aim an novelty of the study are explained. Nonetheless, some corrections are necessary before accepting the text for publishing, as listed below. 

- The manuscript focuses mostly on maize, this should ve highlighted in the title of the ms.

The texts has a rather narrow context, therefore, I suggest changing the term "review" to "mini-review" in the text. 

- Please provide the full botanical name of the species when first mentioned, including the initials of the author

- Citing style in the text, please follow the MDPI format

- Names of genera in Italics

- The description of columns in table on page 5 should be improved. 

- The text is focused on maize, yet for some reasons the authors refer to tobacco in the Conclusions. This is confusing.

- Captions to box 1 and 2 should be placed before the Conclusions. I also do not find it appropriate to provide such a long description to a figure. This should rather be an element of the main body of the text.

Author Response

Response to Reviewer 1 Comments

Point 1- The manuscript focuses mostly on maize, this should ve highlighted in the title of the ms.

Response 1 - No change for two reasons. First, although we do emphasize maize (for reasons given in the Introduction), we also comment on tobacco, Arabidopsis, and animals.  Second, if we specify maize in the title, readers interested in other plants (and animals) may think that there is nothing here for them and not take the time to find that their wider interests are actually addressed.

Point 2 - The texts has a rather narrow context, therefore, I suggest changing the term "review" to "mini-review" in the text. 

Response 2- No change again for two reasons. First, we were invited to submit a “review”, not a minireview.  Second, the length of the text seems to exceed what we expect in a minireview. Although we clearly state that the only species for which both ROS and glycation damage to organellar DNAs during development has been reported is maize, the reviewer evidently thinks there are other species for which such data are published and could be included in our coverage. There are no others, to our knowledge, so that the “narrow context” is unavoidable. However, if the editor decides to require that the category be changed to “minireview” we would (begrudgingly) accept that decision.

Point 3- Please provide the full botanical name of the species when first mentioned, including the initials of the author

Response 3 - Modified as suggested On page 2, line 80, we now write Arabidopsis thaliana L.

Point 4- Citing style in the text, please follow the MDPI format.

Response 4- Modified throughout the manuscript.

Point 5 - Names of genera in Italics

Response 5- Arabidopsis is the common name for Arabidopsis thaliana; therefore, it does not need to be italicized. The only time it needs to be in italics is when given as either Arabidopsis thaliana or A. thaliana.

Point 6 - The description of columns in table on page 5 should be improved. 

Response 6 - We have modified the title and columns for Table 1.  We think these are improvements.

Point 7 - The text is focused on maize, yet for some reasons the authors refer to tobacco in the Conclusions. This is confusing.

Response 7 - We have altered the wording on page 3 lines 126-129 as follows.

Original: Maize was chosen because (i) grasses (such as maize) have a larger leaf meristem than do most angiosperms, including Arabidopsis, facilitating biochemical measurements at all stages of leaf development;

New version: Maize was chosen because (i) monocots (such as maize) have a larger leaf meristem than do most dicots, including Arabidopsis and tobacco (Nicotiana tabacum L.), facilitating biochemical measurements at all stages of leaf development;

Furthermore, the comparison of maize to tobacco in the Concluding remarks illustrates the extremes in the way in which plant species deal with the problem of damage to orgDNA during normal plant development (see point 1, above).  This distinction should prompt future research in this newly-discovered area of glycation damage to organellar DNA as a complement to well-studied ROS damage.  

Point 8 -Captions to box 1 and 2 should be placed before the Conclusions. I also do not find it appropriate to provide such a long description to a figure. This should rather be an element of the main body of the text.

Response 8 - We have now done this with the captions on pages 8 and 9. 

As for the length of the description for Figure 1, we feel that these words are essential for a reader wishing to delve into how we arrived at the rather simple presentation in Figure 1.  We tried to present the “big picture” as simply as possible in the figure for those not wishing to delve into the details.

Reviewer 2 Report

Dear Arnold Bendich and colleagues,

here are my comments on your review describing oxidative and glycation damage in growing maize.

Overall, the review is interesting, well written and informativ.

p1, l35 should it say specific plant parts rather than plant development as the examples refer to the entire plant and the leaf?

p2 l75 should there be references at the end of the paragraph ...by nuclear-encoded, ROS-responsive genes? Are there examples of such genes? If so, inclusion of an example may illustrate the concept better.

p3, l114 here the concept of DNA abandonment appears for the first time and should be briefly introduced to the reader

p5 Table 1 please explain all abbreviations (ie 8-oxoG, MGO, AGE) in the legend

p5 l162 please give here also the full name of 8-oxoG

p6 section 6 please move the section195-211 on the background of glycation to the top of the paragraph before introducing the three bullet points as this allows the reader to become familiar with the concept of glycation. Please explain here briefly how AGE products form via the Maillard & Amadori reactions. Are there AGE receptors in plants?

p6 l197 I think it should say aldehyde group rather than carbonyl group as the aldehyde is the reactive part.

p8 l272 please introduce N2-acetyl-2´-deoxyguanosine to the reader

p8 l 295 is it know whether or how plants might sense glycation products in the nucleus?

p9 l334 how does persistent ssDNA alter the redox status - more information here please. Idea: could it be the release of fragmented orgDNA into the cytoplasm that is the signal?. It may also be worth mentioning that leafs are disposable in many plants. Hence, they can cope for a while with fragmnented orgDNA using long-living RNAs

Author Response

Response to Reviewer 2 Comments

Point 1- p1, l35 should it say specific plant parts rather than plant development as the examples refer to the entire plant and the leaf?

Response 1- No change needed here because the objective in previous research was to investigate the effects of ”perturbation” to an already developed plant tissue, but not the effects of that    perturbation during  the transition from stem cell to differentiated cell. Our intent was is to investigate the “normal” course of tissue development.

Point 2 - p2 l75 should there be references at the end of the paragraph ...by nuclear-encoded, ROS-responsive genes? Are there examples of such genes? If so, inclusion of an example may illustrate the concept better.

Response 2 - No change needed here because in all plants and animals studied to date, the genes carried in   organellar genomes are not ones for either DNA replication or repair. Later we do mention the  signaling between organelle and nucleus, and that hydrogen peroxide is the leading candidate for a signaling molecule, but despite decades of work the only ideas we found are still rather vague and beyond our scope in this review. We do not wish to speculate further on this matter.

Point 3- p3, l114 here the concept of DNA abandonment appears for the first time and should be briefly introduced to the reader.

Response 3 - We do not wish to add additional words here for these reasons. The concept of DNA abandonment is described in the cited Refs [41-43].  One expects damage to DNA to be repaired to avoid deleterious consequence of defective DNA sequences.  Although damaged nuclear DNA is normally repaired, orgDNA damage is frequently not repaired (it is abandoned), perhaps because its copy number is much higher than the two copies in a diploid nucleus, but the integrity of orgDNA varies among plant species for a variety of reasons discussed in Refs [41-43].  Further description of this controversial issue would require many words and is beyond the scope of this review.

Point 4 - p5 Table 1 please explain all abbreviations (ie 8-oxoG, MGO, AGE) in the legend

Response 4-  We added full names for the abbreviations to Table 1 footnotes (page 5, lines 158-159).

Point 5 - p5 l162 please give here also the full name of 8-oxoG.

Response 5 - No change because 8-oxoG is already defined in line 103 of page 3.

Point 6- p6 section 6 please move the section195-211 on the background of glycation to the top of the paragraph before introducing the three bullet points as this allows the reader to become familiar with the concept of glycation. Please explain here briefly how AGE products form via the Maillard & Amadori reactions. Are there AGE receptors in plants?

Response 6 - We moved the section with the three bullet points, as suggested.

No change because the details are quite complex and cannot be explained briefly. However, the  details about the formation of AGE products by Maillard reactions are found in references 49 and 50 on line 192 of page 6.

We cannot provide what the reviewer wants because AGE receptors in plants are still not known. In animals, AGE receptors are transmembrane receptors of the immunoglobulin super family [Shumilina et al., Int J Mol Sci 2019, 20 (9)].

Point 7 - p6 l197 I think it should say aldehyde group rather than carbonyl group as the aldehyde is the reactive part.

Response 7 - No change because reactive carbonyls is the correct term here. Note that one of these is a ketone in methyl glyoxal (CH3-CO-CHO) and fructose.

Point 8 - p8 l272 please introduce N2-acetyl-2´-deoxyguanosine to the reader.

Response 8 - This change has now been made (page 8, lines 275-278).

“MGO caused the conversion of 2'-deoxyguanosine to a DNA adduct, an N2-acetyl derivative of guanine (N2-acetyl-2'-deoxyguanosine), demonstrating their synergistic damaging effects on DNA [64]”.

Point 9 - p8 l 295 is it know whether or how plants might sense glycation products in the nucleus?

Response 9 - No, not to our knowledge. Thus, no change.

Point 10 - p9 l334 how does persistent ssDNA alter the redox status - more information here please. Idea: could it be the release of fragmented orgDNA into the cytoplasm that is the signal?. It may also be worth mentioning that leafs are disposable in many plants. Hence, they can cope for a while with fragmnented orgDNA using long-living RNAs

Response 10 - Nothing added here because we are unaware of firm data concerning the specific signal.  Although the ideas of the reviewer are plausible, we do not wish to add further speculation to our already speculative scenario.

Reviewer 3 Report

antioxidants-2313697-peer-review-v1

Authors presenting an interesting information, however, in several parts sections need to be extended and enriched with more examples.

Ln5: If I am not wrong, the formal name of the country is USA or United States of America.

All around the text, references do not need to be in bold.

Ln49: Please, correct to [5-7]. Follow the same for the other cases in the current manuscript.

Some "parasite" symbols appear in the figures and tables. Please, check. Example in Fig. 1, after ROS is appearing "?"

In my opinion table 1 can be easily summarized as text in few lines. Maybe authors will be considered to have a more complex table, with more examples, better organized?

Ln185-193: Maybe this information can be presented as diagram?

In several places authors have copy-past some information from the referring sources.

PLEASE, check entire manuscript with appropriate plagiarism software to avoid further confusion and legal problems. And when you refer to other work, this cannot be referred as "we studied". Please, see Ln 246. Entire manuscript needs to be scanned again for a similar problem.

Ln294-308: This section needs to be extended.

Please, check reference list for accuracy according with the recommendations for the style from the journal. Some corrections need to be done. (Ln 394, 1831 (1), 186-200, and Ln. 404, 2019, 8 (4):105. What is correct? ","or ":"?

Author Response

Response to Reviewer 3 Comments

Point 1- Ln5: If I am not wrong, the formal name of the country is USA or United States of America.

All around the text, references do not need to be in bold.

Response 1- Corrected. We have unbolded the references throughout the manuscript.

Point 2- Ln49: Please, correct to [5-7]. Follow the same for the other cases in the current manuscript.

Response 2 - We have done this throughout the manuscript (for e.g. Page 2, line 49).

Point 3 - Some "parasite" symbols appear in the figures and tables. Please, check. Example in Fig. 1, after ROS is appearing "?"

Response 3 - We did not notice these symbols on our computers. It may be a software problem.

Point 4 - In my opinion table 1 can be easily summarized as text in few lines. Maybe authors will be considered to have a more complex table, with more examples, better organized?

Response 4 - We cannot summarize in the way the reviewer suggests and wish to retain table 1.

Point 5- Ln185-193: Maybe this information can be presented as diagram?

Response 5 - We are unable to do what the reviewer requests and wish to retain the text as it is.

Point 6 - In several places authors have copy-past some information from the referring sources.

PLEASE, check entire manuscript with appropriate plagiarism software to avoid further confusion and legal problems. And when you refer to other work, this cannot be referred as "we studied". Please, see Ln 246. Entire manuscript needs to be scanned again for a similar problem.

Response 6 - We made the appropriate changes throughout the manuscript. For e.g. please see page 7, lines 251-253.

Point 7 - Ln294-308: This section needs to be extended.

Response 7 - Since the meaning of this very short sentence it is unclear, we have not added any new words. In this paragraph we address a controversial issue identified long ago, and we still have no unifying model that applies to all plants.

Point 8 - Please, check reference list for accuracy according with the recommendations for the style from the journal. Some corrections need to be done. (Ln 394, 1831 (1), 186-200, and Ln. 404, 2019, 8 (4):105. What is correct? ","or ":"?

Response 8 - We have redone references and think they are now in proper form. “,” is correct (page 11, line 410).

Round 2

Reviewer 1 Report

The authors replied to all my comments, although I still consider the title misleading and non-informative as it covers a very narrow area of research. 

Reviewer 3 Report

In my opinion authors have cover the question raised in the first round of reviewing and paper can be suggested for publication